# New Approach for Enhancing Survival in Glioblastoma Patients: A Longitudinal Pilot Study on Integrative Oncology

**DOI:** 10.3390/cancers17142321

**Published:** 2025-07-12

**Authors:** Massimo Bonucci, Maria Pia Fuggetta, Lorenzo Anelli, Diana Giannarelli, Carla Fiorentini, Giampietro Ravagnan

**Affiliations:** 1Association for Research on Integrative Oncology Therapies (ARTOI) Foundation, 00165 Rome, Italy; lorenzoanelli@me.com (L.A.); carla.fiorentini@artoi.it (C.F.); 2Institute of Translational Pharmacology, National Research Council of Italy (CNR), 00133 Rome, Italy; mariapia.fuggetta@ift.cnr.it (M.P.F.); ravagnangp@gmail.com (G.R.); 3Facility of Epidemiology & Biostatistics, Policlinic University Foundation “A. Gemelli” IRCCS, 00136 Rome, Italy; diana.giannarelli@gmail.com

**Keywords:** glioblastoma, STUPP protocol, methylation, integrative oncology, herbal medicine, polydatin, curcumin, *Boswellia serrata*

## Abstract

GBM has a very poor prognosis, even in the short term, because it is one of the most aggressive malignancies and responds very poorly to any medical treatments. This work stems from the need to find ways to improve the therapeutic response to glioblastoma. In a group of 72 patients with primary glioblastoma, those who used natural substances such as polydatin, curcumin, and *Boswellia serrata* in combination with the STUPP radiochemotherapy treatment (60 patients) showed improved survival (from 13.3 months to 25 months of median survival). This study, if further supported by in-depth studies, opens the way for new therapeutic combinations.

## 1. Introduction

Glioblastoma multiforme (GBM), mainly characterized histologically by vascular proliferation and palisade necrosis, is the most common and aggressive type of brain tumor in adults, with a poor prognosis and recurrence rates more frequently [1]. GBM accounts for approximately 50–60% of adult astrocytomas and 12–15% of intracranial neoplasms. The median age at diagnosis is 64 years, with an estimated incidence of 2 to 3 cases per 100,000 individuals in Europe and North America [2]. GBM survival is lower in adult patients than in younger patients [3]. According to the 2021 WHO CNS tumor classification, GBM diagnosis requires molecular criteria including IDH-wildtype and H3-wildtype status [4]. This classification allows histologically lower-grade astrocytomas with aggressive molecular features to be categorized as GBM, IDH-wildtype (WHO grade 4) [5]. The absence of IDH mutations is associated with significantly poorer prognosis.

GBM development involves multiple genetic alterations. Mutations in tumor suppressor genes such as p53, HIC-1, p16, PTEN, and 22q/19q loss of heterozygosity (LOH) are frequently observed, alongside activation of oncogenes like CDK4, MDM2, EGFR, and mutations in IDH isoforms [6]. Many glioblastomas display subclonal amplifications of receptor tyrosine kinases (RTKs) such as MET, EGFR, and PDGFRA within the same tumor [7]. Telomerase reverse transcriptase (TERT) promoter mutations—found in up to 80% of IDH-wildtype and 28% of IDH-mutant glioblastomas—enable telomere maintenance and support cellular immortality [8,9,10]. These mutations are prognostically relevant and linked to worse outcomes [11].

Epigenetic markers such as MGMT promoter methylation are crucial in guiding treatment decisions. Methylation status predicts Temozolomide (TMZ) response, with approximately 35–45% of patients carrying this favorable marker [12,13]. However, the genetic and epigenetic heterogeneity of GBM, often present even within a single tumor, limits the effectiveness of any single-target therapy. Angiogenesis is a central pathophysiological feature of GBM. Vascular endothelial growth factor (VEGF) and other cytokines promote abnormal vasculature characterized by hyperpermeability, enlarged vessel diameter, and thickened basement membranes [14]. Histologically, GBM exhibits cellular pleomorphism, necrosis, and high mitotic activity—hallmarks underlying the term “multiform” [15]. Tumor infiltration into surrounding brain tissue complicates complete surgical resection. Glioma stem cells, resistant to therapies, can disseminate to distant brain regions, contributing to recurrence [16,17]. Even minimal residual disease can proliferate, leading to tumor regrowth [18,19]. Additionally, GBM is associated with increased arachidonic acid metabolism, which promotes peritumoral cerebral edema and tumor progression [20].

The current standard of care includes surgical resection [21], radiotherapy, and adjuvant TMZ, an alkylating agent derived from dacarbazine [22]. TMZ improves progression-free survival (6.9 vs. 5 months) and overall survival (14.6 vs. 12.1 months) [23]. Combining chemotherapy with radiotherapy provides superior results compared to radiotherapy alone [24]. All new studies on radiotherapy focus on the value of re-irradiating recurrences using conventional or stereotactic radiotherapy. Additional agents such as bevacizumab and other alkylators (e.g., carmustine, lomustine, nimustine, fotemustine) are also used [25,26,27,28]. Nonetheless, many novel therapies, including targeted drugs, have failed to demonstrate significant clinical benefit [29]. One promising agent, regorafenib, showed survival benefits in relapsed GBM, increasing survival by about seven months in a University of Padua study [30]. Treatment-associated side effects remain a challenge. Radiotherapy frequently requires steroids to reduce cerebral edema and intracranial pressure [31], but glucocorticoids have well-known adverse effects [32]. Given the modest efficacy and toxicity of current therapies, there is growing interest in identifying novel agents with better tolerability and improved outcomes.

Natural compounds offer potential benefits due to their pharmacological activities and relatively low toxicity. Many patients turn to traditional herbal medicine to support immune function and counteract tumor progression [25,26,33]. In vitro studies have shown that some natural substances inhibit proliferation, induce apoptosis, and affect telomerase activity in GBM cells [34]. Extracts from *Angelica sinensis* [35], inositol hexaphosphate [36], *Nigella sativa* [37], *Scutellaria baicalensis* [38], *Artemisia argyi* [39], and *Tithonia diversifolia* [40] show pro-apoptotic effects. Other extracts, including *Garcinia mangostana* [41] and *Balanites aegyptiaca* [42], produce ROS, while *Panax ginseng* (Rg3) [43], *Cannabis sativa* [44], *Vitis vinifera* [45], and magnolol [46] display antiangiogenic effects. Furthermore, *Camellia sinensis* [47], *Tinospora cordifolia* [48], resveratrol [49], and curcumin [11] exhibit antimetastatic activity. Polyphenols are plant-derived compounds with antioxidant, anti-inflammatory, and antitumor properties [50,51,52,53,54]. Curcumin (CUR), the major component of *Curcuma longa*, has demonstrated anticancer activity across multiple tumor types, including GBM [55]. It regulates various signaling pathways, reducing proliferation and promoting apoptosis [56]. Polydatin (PD), a glycosylated form of resveratrol derived from *Polygonum cuspidatum*, also exhibits antioxidant, anti-inflammatory, and antiproliferative effects [57,58,59,60,61]. In GBM models, PD inhibits proliferation, migration, invasion, and stemness while promoting apoptosis [62]. Both CUR and PD are widely used in nutraceuticals and pharmaceuticals. However, no clinical studies have evaluated their combined use with standard GBM therapies [63]. Both agents modulate key factors such as VEGF, FGF, PDGF, HIF-1, TNF-α, MMP-9, and COX-2, reduce proinflammatory cytokines (e.g., IL-1, IL-6, IL-8, IL-17), and affect apoptosis-related proteins like Bax, Bcl-2, and p53.

This longitudinal pilot study, though non-randomized and non-blinded, aims to assess whether the combined use of CUR and PD with standard therapies improves survival and quality of life in patients with GBM (diagnosed only by histological examination and MGMT promoter molecular study alone, with a classification of WHO grade 4 glioma). Additionally, *Boswellia serrata* (BS) extract was introduced for its anti-edematous effects and potential to reduce glucocorticoid dependence [64]. BS, long used in Ayurvedic medicine [65], exhibits antitumor activity through downregulation of inflammation, proliferation, invasion, and angiogenesis markers [66,67]. It targets NF-κB, COX-2, and LOX-5 pathways [68], although data on its role in brain tumors remain limited [69].

While the mechanisms of many natural products are well understood, their synergistic potential when combined in a clinical setting remains largely unexplored [70]. This study, for the first time, investigates whether multi-compound natural therapy provides additive benefits in the primary GBM treatment. Specifically, it aims to determine whether the addition of curcumin, polydatin, and *Boswellia serrata* to standard therapy can improve survival and quality of life in patients with IDH-wildtype glioblastoma. The study also examines whether these natural compounds can reduce the need for corticosteroids and offer complementary therapeutic effects that have not yet been explored in combination.

## 2. Materials and Methods

### 2.1. Patients

Patients with newly diagnosed, only histologically confirmed, glioblastoma multiforme, and by MGMT promoter molecular study (WHO grade 4 glioma) whose lesions were classified before the advent of molecular biology IDH mutation, who underwent surgery (or simple biopsy), radiotherapy, and chemotherapy with Temozolomide (TMZ) were enrolled in this longitudinal pilot study after providing their written informed consent. Patients came to the outpatient clinic after initial treatment in the public hospital. Their status was expected to be good, KPS (Karnofsky Performance Status) > 70, and they already had a histologic diagnosis. They had no sensory/motor deficits that would limit their ambulation, speech, swallowing, and/or chewing. The study adhered to the Helsinki Declaration and received approval from the institutional Ethics Committee (Ethics Committee Lazio 1—Rome (prot. n° 336/CE Lazio 1. 1 March 2018). A total of 72 patients with primary GBM were included. This group, on a voluntary basis, was given the opportunity to take natural substances in conjunction with the STUPP protocol. The whole group was monitored and treated from September 2012 to March 2017 at the Oncology Unit of a Roman Hospital. We did not consider patients who presented with GBM recurrence in this study. The choice not to introduce patients with recurrence was made because of the small number who presented and to have a more homogeneous sample at the beginning, although we knew that the final data might have less statistical value.

Inclusion criteria were as follows:Males and females;Age over 18 years old;Patients able to swallow and/or chew tablets and/or liquids;KarnofskyPS > 70;Adequate hematologic, liver, and kidney function (results of laboratory conducted within 7 days after the start of the study: hemoglobin > 9.0 g/dL; WBC > 3.0 × 109/L; absolute neutrophil count (ANC) > 1500/mm^3^ without transfusion or stimulation with hematopoietic growth factors; platelet count 100,000/μL; total bilirubin < 1.5, the upper limit of normal; alanine transferase (ALT) and aspartate transferase (AST) < 3, the upper limit of normal; serum creatinine < 1.5, the upper limit of normal;Patients able to understand and give their own informed consent;Histological confirmation of glioblastoma (WHO grade 4);Radiographic evidence of brain cancer;All patients diagnosed radiologically with a probable neoplastic lesion of cerebral origin who underwent surgical biopsy for histological diagnosis;Patients who underwent surgery (total, partial, extensive, or simple biopsy) for neoplasm removal;In all cases, biological parameters and the assessment of the lesion’s methylation status (O^6^-methylguanine-DNA methyltransferase (MGMT) promoter methylation) were studied.

Exclusion criteria were as follows:Previous chemotherapy and radiotherapy treatment for GBM;Known hypersensitivity to any of the products used in the study;Karnofsky Performance Status (KPS) < 70;Having taken systemic antiblastic therapy, including cytotoxic therapy, signal transduction inhibitors, immunotherapy, and/or hormone therapy within 4 weeks before the start of the study;Severe comorbidities (heart disease, recent stroke, kidney failure (moderate/severe), autoimmune disease);Secondary malignancies;Intake of natural products containing curcuminoids, resveratrol, and *Boswellia serrata* in the 6 months prior to the start of the study.

### 2.2. Radiochemotherapy and Complementary Treatment

Patients who received radiotherapy (radiation treatment) underwent 30 sessions on the whole brain. The radiotherapy provided 59.4 Gy in 30 daily fractions (1.8–2 Gy for each fraction), totaling 6 weeks of treatment. It commenced within 28 to 30 days following surgery, coinciding with the expected healing process. In conjunction, Temozolomide (TMZ) chemotherapy was administered daily, starting with the radiotherapy and continuing until its completion. When feasible, TMZ was continued for an additional month after the treatment’s conclusion, following a 4-week chemotherapy-free interval. Patients received up to six cycles of maintenance (adjuvant phase) TMZ (150–200 mg/sqm for 5 days during each 28-day cycle). This regimen was given daily for 5 days every 28 days. Patients continued therapy with steroids at the initial prescribed dose; any modifications were evaluated during clinical visits, considering the edema assessed on MRI and the patients’ medical condition.

Of the 72 patients who participated in the study, 7 (9.7%) after the first visit, where it was explained how the integrative treatment was to be followed and the type of nutrition to be followed, had never received integrative therapy, and an additional 5 (6.9%) were not compliant with the IT protocol, meaning they did not consistently adhere to medical prescriptions.

Integrative treatment (IT) was achieved using a pharmaceutical formulation known as Composition (Table 1). This formulation combines polydatin PD (CAS number 27208-80-6) and curcumin CUR. Curcumin CUR consists of curcumin I, curcumin II, and curcumin III. The Composition used for IT contained approximately 2 mg to 4 mg of PD and 2 mg to 5 mg of CUR per kilogram of body weight, depending on the patient’s physical condition. The pharmaceutical formulation of Composition (gel and/or mouth-soluble tablet) was administered daily for at least six weeks during the radiotherapy treatment cycle and preferably for the remainder of the patient’s life. In the formulation, the active compounds are mixed with a carrier that possesses the necessary binding properties in suitable proportions.

Two administration intervals were identified, one related to the acute phase and another pertaining to the maintenance phase. During the acute phase, which can last up to one year, patients received at least 500 mg of the Composition daily. In the maintenance phase, which can extend for the remainder of the patient’s life, patients received 300 mg of the Composition daily. Patients commenced treatment with the Composition from the start of their diagnosis. Some patients, however, opted to initiate IT during various phases of conventional treatment.

The Composition has been administered alongside a compound of *Boswellia serrata* extract that possesses anti-edema activity. This innovative phytosome-based delivery form of boswellic acids is provided at a dosage range of 1.8 to 2.4 g per day (according to body weight (< or >70 kg) and the degree of perilesional encephalic edema, as assessed on MRI images (from G1 to G3 sec CTCAE v5.0). During treatment with Temozolomide, patients underwent thorough monitoring of all five vital parameters (hematological tests) as well as instrumental examinations (CT and MRI) to evaluate disease progression and/or edema. No major changes in the above parameters were noticed during the administration. If hematological parameters (white blood cells, red blood cells, platelets, liver enzymes (from G1 to G3–4 sec CTCAE v5.0)) were altered during Temozolomide treatment, in cases where conventional treatments (such as growth factors and cortisone) have not been able to provide adequate coverage, certain natural substances, such as Tamarix gallica extracts (for enhancing red blood cells), melatonin (for low platelets), and glutathione (for high liver enzymes), were administered to mitigate the effects. The instrumental examinations were repeated every three months following the completion of radiotherapy for a minimum of two years, subsequently shifting to every six months throughout the study period. This examination served to assess disease progression and the presence of brain edema. Both cortisone and a higher dosage of *Boswellia serrata* were used to reduce cerebral edema. Patients adhered to a specialized nutrition regimen recommended by the ARTOI Foundation (Table 2). The diet excludes red and white meat, milk and dairy products, simple and complex sugars, and foods containing polyamines, while the intake of fruits and soy is reduced.

### 2.3. Statistical Analysis

Overall survival was measured from the date of definitive glioblastoma diagnosis or the initiation of IT, as appropriate, to the date of death or the most recent follow-up available. Survival curves were estimated using the Kaplan–Meier method and compared using the log-rank test. Median survival time and corresponding 95% confidence intervals were reported. A Cox proportional hazards model was implemented to adjust the effect of IT based on demographic and clinical characteristics; univariate hazard ratios (HRs) and their 95% confidence intervals (95% CIs) are reported. Only factors with a *p*-value < 0.10 were considered in the multivariable approach; selection was based on the Wald statistics. Baseline comparisons were analyzed using the χ^2^ test. A *p*-value of less than 0.05 was deemed statistically significant. All statistical analyses were performed using SPSS version 21.0. (IBM SPSS Statistics for Windows (Version 21.0), IBM Corp., Armonk, NY, USA).

## 3. Results

The statistical analysis, in our work, shows that factors significantly associated with improved outcomes include surgery (*p* = 0.003), radiotherapy (*p* < 0.001), and adherence to IT (*p* = 0.001). When we use a multivariate approach, methylation also becomes significant (*p* = 0.038), and the other three factors (radiotherapy, adherence, and surgery) also remain significant. The hazard ratios (HRs) are all less than 1.00, meaning that radical surgery, radiotherapy, and adherence decrease the risk of death significantly and independently of each other. Age, gender, chemotherapy, and steroid use do not impact OS (Table 3). As shown in Table 4, the patient population consisted of 42 males (58%) and 30 females (42%). The average age was 57 years (range: 18–85 years). Regarding integrative treatment (IT), a large proportion (83.3%) were highly adherent, 6.9% received IT but were not adherent, and 9.7% never participated in it. Lastly, corticosteroid use was divided; 43.1% received it at the start of the study, while 56.9% did not.

Patients with GBM visited our center for the first time, on average, 3.9 months after diagnosis, with a range of 10 days to 14 months. At the time of their initial visit, 31 patients (43.1%) had received corticosteroids.

This work was supported by the Oncology Ethics Committee of the Lazio Region (Prot. N.336/CE LAZIO 1. Rome 1 March 2018).

Concerning the treatment type, Table 5 shows that among the 72 patients, surgical intervention was prevalent, with 91.7% undergoing some form of surgery. Among these patients, 66 (91.7%) had undergone surgery, with gross total resection achieved in 28 cases (38.9%), while 52.8% underwent either a partial resection or extensive biopsy. Six patients had only a simple single biopsy for diagnosis (8.3%). A total of 69 patients (95.8%) had received radiotherapy. Six patients (8.3%) had not undergone any chemotherapy, while the other sixty-six received it. Fifty-three patients (73.6%) had received only one line of chemotherapy. Of these, 52 patients were treated with TMZ, and only 1 received Nimotuzumab plus Vinorelbine. Eleven patients (15.3%) had a second treatment, primarily consisting of fotemustine. Two patients (2.8%) underwent a third treatment, which included one BCNU Carmustine combined with a PCV-conjugated pneumococcal vaccine and one rituximab.

Fifty-three patients underwent chemotherapy. In first-line chemotherapy, Temozolomide (TMZ) was the most common choice, at 72.2%. Alternative regimens, such as Nimotuzumab combined with vinorelbine, were rarely utilized, accounting for only 1.4%. Second-line chemotherapy was less prevalent, with just 15.3% receiving fotemustine. Third-line chemotherapy was even less common, with only two patients receiving additional treatment.

As shown in Table 6, of the 72 cases examined, 25 showed methylation, while 47 tested negative. Of the 28 cases with complete excision, 11 were positive and 17 were negative. The table also includes the ten long-survival cases, six of which showed methylation and four of which were negative.

The treatment approach for this patient group was highly standardized, primarily utilizing radiotherapy and Temozolomide (TMZ) as the main therapeutic strategies. Although most patients underwent surgery, gross total resection was achieved in only about a third of cases. The use of second- and third-line chemotherapy was less frequent, indicating that treatment intensity decreased as the disease progressed. Furthermore, integrative treatments were widely embraced, reflecting a trend toward supportive care. The variability in corticosteroid use indicates differences in strategies for managing symptoms. This data illustrates a typical treatment pathway for patients, emphasizing the importance of multimodal therapy while showing that aggressive interventions become less common in later stages.

As shown in Table 7, the median overall survival (OS) for the 65 patients who received integrative treatment was 16.3 months (95% confidence interval [CI]: 0.2–32.4). The series of 60 patients who demonstrated high adherence to the therapy achieved a median OS of 25.4 months (95% CI: 8.3–42.5). The one-year survival rate for the entire series was 53.1%, with a 55.4% rate for the 65 patients treated. The series of 60 patients who exhibited high adherence achieved a one-year survival rate of 59.0%.


**Additional Survival Data**


**Survivors at study conclusion**: 10 patients remained alive at the end of the study.**Long-term survivors**: Approximately 12% of total patients (increasing to >16% in the IT protocol group) survived beyond 60 months.**Median survival update**: For the sample of survivors, median survival now exceeds 55 months.


**Comparison with Literature**



**Source**

**5-Year Survival Rate**
Current study (all patients)16.0%Current study (complete surgery + high adherence)34.0%Literature references3–7%

No significant differences in survival were observed based on gender, age, or the number of previous chemotherapy treatments (Table 8). The main prognostic factors for glioblastoma (GBM) are age, performance status, histologic grade, and the presence of specific molecular markers, such as MGMT methylation, IDH1/2 mutations, 1p19q codeletion, and EGFR overexpression, according to a new classification. In our study, notably, the one-year survival rates were 51.9% for males and 59.1% for females. For younger patients (<57 years), the one-year survival rates were 53.8%, while older patients had rates of 57.0%. Patients who received only one chemotherapy treatment had a one-year survival rate of 58.0%, compared to 56.3% for those who underwent two or three treatments. Additionally, patients who received corticosteroids had a one-year survival rate of 58.7%, whereas those who did not receive them had a rate of 50.6%. The overall average survival across the entire series varied depending on the extent of surgical intervention. Patients who did not undergo surgery (only simple biopsy for diagnosis) had a one-year survival rate of 16.6%, those who had gross total surgery achieved a one-year survival rate of 75.0%, and patients who only had extensive biopsy/partial surgery had a one-year survival rate of 57.9%. The median overall survival for this group from the time of diagnosis was 34.4 months (95% CI: 18.1–40.8), with a one-year survival rate of 82.4% and a two-year survival rate of 54.2%.

When the side effects were considered (Table 9) (evaluated with a CTCAE v5.0 scale), unexpected results were noted with IT, including a reduction in the adverse effects of radiotherapy. Specifically, there was a decrease in post-radiation edema (seen primarily with MRI), and there were no special events related to the use of glucocorticoids. Using *Boswellia serrata* extract as a supplement, and post-radiation asthenia was only verbally assessed in patients, the latter of which has not yet been published. The platelet count among the 60 patients was normal, or not reduced if they used melatonin, as were the white and red blood cell counts. These findings allowed chemotherapy treatment to continue without interruption for all desired cycles.

## 4. Discussion

Glioblastoma (IDH-wildtype, WHO 2021), a highly aggressive neoplastic disease, is associated with a dismal five-year prognosis [1]. Diagnosis typically occurs when the lesion is already extensive, and significant surgical intervention often proves unfeasible due to unfavorable prognostic factors [1]. This condition may result from the progression of low-grade glial lesions that, over time, develop aggressive features through specific mutations and or the acquisition/presence of specific genetic mutations in isocitrate dehydrogenase (IDH mutations, leading to increased malignancy) [6]. GBM carries a poor prognosis, with patients rarely achieving prolonged survival. Even with multimodal therapies such as radiotherapy, chemotherapy, and surgery, the prognosis remains challenging. All new studies on radiotherapy focus on the value of re-irradiating recurrences using conventional or stereotactic radiotherapy [72]. Notably, surgery has been correlated with more favorable outcomes. It is accepted that the surgical radicality in prime GBM resections is a prognostic factor for PFS and OS, while the presence of a residual tumor is not [73]. Literature reports have stated that the median survival for GBM patients after a second resection is between 7 and 12.4 months [74,75].

Currently, concomitant chemotherapy with alkylating agents, particularly Temozolomide, is the standard approach for treating glioblastoma. The median survival rate is approximately 18 months, while the five-year survival rate ranges from 2% to 7% [23]. The various treatments available today, including biological and immunotherapeutic approaches, as well as those involving nanoparticles or radiometabolic strategies, do not seem to alter the survival rates of these patients significantly. However, they often demonstrate a substantial burden of adverse effects that severely restrict their comprehensive application [29]. Given the aggressive nature of glioblastoma, which frequently exhibits resistance to radio- and chemotherapy, along with the adverse effects associated with these modalities, there is an urgent need for a novel and appropriate alternative or adjuvant treatment for glioblastoma.

Over the past few decades, there has been a growing interest in using herbal and natural compounds, or their derivatives, which tend to be less toxic, for various therapeutic purposes, particularly in cancer treatment. Natural substances have consistently demonstrated efficacy in promoting antineoplastic activity against high-grade glial cells in both in vitro and in vivo studies. Furthermore, in the management of glioblastoma, over 50% of patients utilize complementary and alternative approaches, with herbal therapies being the most commonly employed empirically [33]. Glucosidic stilbenoids, such as resveratrol and curcumin, are widely utilized in the pharmaceutical industry, either alone or in combination. The US Patent 2009/0047371 describes the application of resveratrol and curcumin compositions for treating prostate tumors and inflammatory diseases, including psoriasis and other skin conditions. However, there is currently no evidence suggesting that the combination of PO and CUR can effectively treat tumors in the central nervous system (CNS). The synergistic effects of curcumin alongside radiotherapy and chemotherapy illustrate its potential in treating glioblastoma multiforme (GBM). Curcumin stands out as an exemplary natural pharmaceutical compound capable of crossing the blood–brain barrier (BBB). Additionally, its lipophilic nature promotes favorable absorption, availability, and retention within the CNS [76]. Numerous scientific studies have explored the various pharmacological effects of curcumin, which include antimicrobial, anti-inflammatory, antioxidant, and notably, anticancer properties [55]. In this context, curcumin has been recognized as an effective antitumor agent against GBM [11]. The anticancer properties of curcumin encompass modulation of cell proliferation, induction of apoptosis, inhibition of angiogenesis, induction of autophagy, stimulation of the immune response, and inhibition of cell invasion and metastasis [56]. Despite its potential therapeutic benefits, the use of curcumin is limited due to its low intestinal absorption and pharmacokinetics [77,78]. After repeated administration of curcumin in humans, the blood serum concentration peaked at approximately 2 μM [79], which may be insufficient for effective anticancer action. Curcumin exhibits a remarkably low toxicity profile, and numerous clinical studies have demonstrated its well-tolerated and safe nature (Clinicaltrials.gov 2019). Polydatin is a polyphenolic compound that acts as a phytoalexin, a naturally occurring substance found in various plant species, including *Polygonum cuspidatum* [57]. Polydatin is the glycosylated form of resveratrol (RSV) and represents the most abundant derivative of resveratrol in nature [58]. Many studies have investigated the beneficial effects of PD/RSV on the human body, highlighting their antioxidant, anti-inflammatory, antitumor, antiviral, neuroprotective, hepatoprotective, and ischemia-preventing activities. Furthermore, the mechanisms of action of PD/RSV have been elucidated [59,60,61]. Specifically, PD can inhibit proliferation, migration, invasion, and stemness in GBM cells while inducing apoptosis. A variety of conventional chemotherapeutic agents are used to treat GBM, including Temozolomide, doxorubicin, paclitaxel, and others. Several studies have shown that resveratrol can enhance the therapeutic efficacy of these chemotherapeutic agents through various mechanisms, which will be discussed in detail below [59]. Resveratrol has been demonstrated to significantly improve the radiosensitivity of cancerous cells in both in vitro and nude mouse models. This effect is attributed to its synergistic anticancer properties, including inhibition of self-renewal and stemness, induction of apoptosis, induction of autophagy, and inhibition of DNA repair [80]. Self-renewal is a fundamental characteristic of stem cells, and this capacity of cancer stem cells (CSCs) is crucial for tumorigenesis and tumor progression [81]. CSCs possess stemness potential, indicating that proliferative cancer cells are continuously renewed through the asymmetric division of CSCs [82,83]. A study involving primary glioblastoma multiforme (GBM) patients who were referred to an oncology outpatient clinic and received standard care (the STUPP protocol), followed by second- and third-line anticancer drugs, was conducted. As part of the adjunctive protocol, herbs such as polydatin, curcumin, *Boswellia serrata*, and nutritional recommendations from the American Society of Clinical Oncology (ASCO) were implemented.

In this observational study conducted at the oncology outpatient clinic at S. Feliciano Hospital in Rome, 72 patients with primary glioblastoma were evaluated over a five-year period (2012–2017). Diagnosis was performed based on histological examination and the presence/absence of MGMT promoter mutation. At that time, the study of the IDH1 gene was not yet standardized (and we have not had the opportunity to reevaluate all cases with the new method). The study was performed without a randomized control group. Patients voluntarily had the opportunity to take natural substances in conjunction with the STUPP protocol. Among them, seven patients did not receive IT therapy, and five either did not adhere to the IT protocol or did so for less than 4 months, while the remaining sixty patients consistently followed the protocol. All patients provided consent for participation and additional treatment. The results obtained from these patients were significant, revealing a median survival of 25 months, with 10 patients surviving beyond 60 months (approximately 12% of the total number of patients, which increases to over 16% in the IT protocol group). Finally, when compared with literature data on glioblastoma, the results of this study show some benefits of utilizing integrative treatment (Table 10). The sample survivors are significant, with a median survival time now exceeding 55 months. The current survival rate is over 25% among those closely adhering to the protocol (60 out of 72 patients). In contrast, the total study population exhibits a survival rate of 16%. The subgroup that underwent gross total surgery had an overall survival rate of 34%, while the international 5-year survival rate ranges from 3 to 7% [1,3,23]. At the conclusion of the study, ten patients remained alive.

This observational study, although it is without a control group and not randomized, aimed to clarify the potential synergistic effects of natural substances when paired with conventional chemotherapy and radiotherapy methods. The study showed that these patients maintained hematological standards that allowed them to receive a broader range of anticancer treatments. The use of a multimodal treatment regimen, which includes the aforementioned innovation, seems to act as a valuable adjunct in extending the survival of patients with brain tumors. Furthermore, this approach seems to ensure a satisfactory well-being for these individuals during the period of treatment. The data presented here appears to be able to contribute a significant improvement in the survival rate of patients undergoing treatment for glioblastoma, as evidenced by their continued existence and better symptomatology.

### Limitations of the Study

Several limitations of the study must be acknowledged. First, the study does not include patients with relapsed glioblastoma. This decision was made due to the small number of patients who presented and the need for a homogeneous group for the study. Second, this was a preliminary investigation that lacked a double-blind, randomized design. Although 72 cases were evaluated, 7 patients did not receive integrative therapy and can be considered a control group. Therefore, the study lacks a proper control group and randomization. Third, the voluntary choice to participate in the study may have involved the emotional sphere of patients and also influenced adherence (in a positive sense), and this influence is better evaluated with an appropriate control group. The research focused on first-onset glial neoplasms. To ensure consistent validation of the treatment with curcumin, polydatin, and Boswellia, the study should be expanded to include patients with glioblastoma relapses, allowing for the evaluation of the effects on reducing or blocking proliferative activity. Preliminary unpublished studies indicate a synergistic effect with Temozolomide, which may lead to the development of a relapse prevention strategy using low-dose Temozolomide in conjunction with curcumin and polydatin. Extensive resection may contribute to some of the observed benefits.

Previous studies involving patients with varying degrees of resection suggest that complete tumor removal may extend survival by approximately 4 to 5 months compared to partial resection [19]. This time is enhanced if there could be a second resection, and the median will be between 7 and 12.4 months [74,75]. However, when considering the total number of survivors, only 16% truly experienced the benefits at the 5-year follow-up. This figure varies significantly in the literature, ranging from two to five times. This discrepancy deserves careful consideration and should encourage us to collaborate on innovative approaches to improve cure rates and ultimately ensure salvation for everyone. We know that our longitudinal, nonrandomized study, in the absence of a true control group, according to the EANO guideline, does not have a good degree of evidence and recommendation (it could be placed in Level C/Good practice point) [84], but it is still a study that highlights a potential aiding action by natural substances that do not interfere with conventional therapy and that seems to give some benefit. The last limitation is that the IDH gene status has not been studied since this testing was not available for all patients at the time of the study’s initiation. It was not possible to re-evaluate all cases with this method due to the impossibility of finding histological preparations of all patients. This data is currently considered critical in the certainty of the diagnosis of glioblastoma. This finding may not fully correlate with the diagnosis of glioblastoma in all cases, but may also contain cases of grade 4 malignant glioma.

## 5. Conclusions

This study suggests that integrative therapy combining natural compounds, specifically polydatin, curcumin, and *Boswellia serrata*, with standard treatment protocols might contribute a potential benefit in prolonging survival in glioblastoma patients, especially those who undergo gross total resection. These natural compounds appear to enhance therapeutic efficacy through various mechanisms, including apoptosis induction, anti-inflammatory effects, and radiosensitization.

However, the small control group size, lack of randomization, the lack of data on IDH mutational status, and the focus on primary glioblastoma limit the generalizability of the results. Future research should address these limitations by conducting larger, randomized controlled trials (RCTs) to confirm the clinical efficacy of this integrative approach. Further studies should also explore the optimal combination and dosing strategies for polydatin, curcumin, and *Boswellia serrata*, as well as their impact on both survival and quality of life.

In conclusion, while the results are promising, robust clinical trials are necessary to establish the safety and effectiveness of these natural adjuncts in glioblastoma treatment.

## Figures and Tables

**Table 1 cancers-17-02321-t001:** Integrative treatment details for patients with glioblastoma multiforme.

Component	Composition	Dosage	Administration Schedule
**Main Composition**			
Polydatin (PD)	CAS number 27208-80-6	2–4 mg per kg of body weight	Daily
Curcumin (CUR)	Mixture of curcumin I, II, and III	2–5 mg per kg of body weight	Daily
**Administration Forms**			
Acute phase (up to 1 year)	PD + CUR (SHERMAN Tree Nutraceuticals—Italian supplement)	500 mg of Composition daily	For at least 6 weeks during radiotherapy and continuing through the acute phase (one year)
Maintenance phase	PD + CUR (SHERMAN Tree Nutraceuticals—Italian supplement)	300 mg of Composition daily	For the rest of the patient’s life
**Pharmaceutical Formulation**			
Delivery method	Gel and/or mouth-soluble tablet	-	-
**Adjunct Treatment**			
*Boswellia serrata* extract	Phytosome-based delivery form of boswellic acids (65%) (Biofarmex—Italian supplement with 70% of ACBA acids)	1.8–2.4 g total dose per day	Daily
**Supportive Treatments**			
For hematological parameters alterations			
	Tamarix gallica extracts (Italian supplement)	20 drops × 3 times a day until normal red blood cells	As needed during Temozolomide treatment
	Melatonin (PRM Factory—Italian supplement)	20 drops until normal platelets	As needed during Temozolomide treatment
	Glutathione (PRM Factory—Italian supplement)	600 mg/day if the liver enzymes are modified	As needed during Temozolomide treatment

**Table 2 cancers-17-02321-t002:** Dietary regimen for patients with glioblastoma multiforme (WHO grade 4 glioma).

Dietary Category	Recommendation	Details/Rationale
**Foods to Exclude**		
Meat	Completely excluded	Both red and white meat
Dairy	Completely excluded	Milk and all dairy products
Sugars	Completely excluded	Both simple and complex sugars
Polyamine-rich foods	Completely excluded	Foods containing high levels of polyamines [71]
Soy products	Completely excluded	Limited consumption recommended
**Foods to Reduce**		
Fruit	Reduce intake	Limited consumption recommended
**Dietary Framework**		
Dietary structure	Specialized nutrition regimen recommended by ARTOI	According to ARTOI guidelines
Implementation timing	Begin at diagnosis and continue throughout treatment	Concurrent with conventional and integrative treatments
**Dietary Rationale**		
Cancer metabolism	Reduce glucose availability	GBM cells are highly dependent on glucose metabolism
Inflammation	Reduce inflammatory dietary components	Support anti-inflammatory effects of integrative treatments
Polyamine pathway	Reduce dietary polyamines	May influence tumor growth [71]
**Monitoring**		
Adherence assessment	During follow-up visits	Part of overall treatment adherence evaluation
Nutritional status	Regular assessment	To prevent malnutrition

**Table 3 cancers-17-02321-t003:** Statistical analysis.

	Univariate HR (95% CI)	Multivariable HR (95% CI)
GENDER		--
Male	1.19 (0.62–2.30) *p* = 0.61	
Female	Ref.	
AGE (in years)	1.01 (0.99–1.04) *p* = 0.36	--
SURGERY		
Biopsy/partial resection	Ref.	Ref.
Radical resection	0.30 (0.130.66) *p* = 0.003	0.39 (0.16–0.94) *p* = 0.036
RADIOTHERAPY		
No	Ref.	Ref.
Yes	0.07 (0.02–0.26) *p* < 0.001	0.04 (0.01–0.22) *p* < 0.001
CHEMOTHERAPY		--
No	Ref.	
Yes	0.95 (0.29–3.11) *p* = 0.94	
CORTICOSTEROIDS USE		--
No	Ref.	
Yes	0.90 (0.47–1.75) *p* = 0.76	
METHYLATION		
No	Ref.	Ref.
Yes	0.48 (0.22–1.02) *p* = 0.058	0.44 (0.20–0.95) *p* = 0.038
ADHERENCE TO IT		
No	Ref.	Ref.
Yes	0.31 (0.15–0.63) *p* = 0.001	0.34 (0.15–0.74) *p* = 0.007

**Table 4 cancers-17-02321-t004:** Demographic and clinical characteristics of patients with glioblastoma WHO grade IV.

Characteristic	Value	Percentage (%)
**Total number of patients**	72	100
**Gender distribution**		
- Male	42	59
- Female	30	41
**Age**		
- Range	18–85 years	
**Corticosteroid treatment**		
- At the beginning of the study	31	43.1
**Average time from diagnosis to first visit**	3.9 months	
- Range	10 days–14 months	
**Adherence to integrative treatment**		
- Never received integrative therapy	7	9.7
- Non-adherent to protocol	5	6.9
- High adherence to protocol	60	83.3

**Table 5 cancers-17-02321-t005:** Treatments received by patients with glioblastoma (WHO grade 4 glioma).

Treatment Type	Number of Patients	Percentage (%)
**Surgical Intervention**		
- Gross total resection	28	38.9
- Partial resection/extensive biopsy	38	52.8
- Only a single biopsy	6	8.3
**Radiotherapy**		
- Received radiotherapy	69	95.8
- Did not receive radiotherapy	3	4.2
**Chemotherapy—First Line**		
- Temozolomide (TMZ)	52	72.2
- Nimotuzumab + Vinorelbine	1	1.4
**Chemotherapy—Second Line**		
- Fotemustine (primarily)	11	15.2
**Chemotherapy—Third Line**		
- BCNU Carmustine + PCV-conjugated pneumococcal vaccine	1	1.4
- Rituximab	1	1.4
**Integrative Treatment (IT)**		
- Received and highly adherent to IT	60	83.3
- Received but not adherent to IT	5	6.9
- Never received IT	7	9.7
**Corticosteroid Treatment**		
- Received at beginning of study	31	43.1
- Did not receive at beginning of study	41	56.9

**Table 6 cancers-17-02321-t006:** Distribution of methylation of cases: totals, survivors, no therapy, and no adherence.

CATEGORY/TREATMENT	TOTAL	METHYLATION	
		YES	NO
TOTAL CASES	72	25	47
TOTAL SURGERY	28	11	17
PARTIAL SURGERY	38	11	26
OLNY BX	6	2	4
SURVIVORS	10	6	4
TOTAL SURGERY	7	4	3
PARTIAL SURGERY	3	2	1
NO THERAPY	7	2	5
TOTAL SURGERY	1	1	-
PARTIAL SURGERY	3	1	2
ONLY BX	3	-	3
NO ADHERENCE	5	2	3
TOTAL SURGERY	1	1	-
PARTIAL SURGERY	4	1	3

**Table 7 cancers-17-02321-t007:** Overall survival results for patients with glioblastoma multiforme who received integrative treatment.

Patient Group	Number of Patients	Median Overall Survival (Months)	95% Confidence Interval	1-Year Survival Rate	2-Year Survival Rate	5-Year Survival Rate
All patients	72	13.3	7.3–19.3	53.1%	-	16.0%
Patients who received integrative treatment	65	16.3	0.2–32.4	55.4%	-	-
Patients with high adherence to integrative treatment	60	25.4	8.3–42.5	59.0%	-	25.0%
Patients who underwent complete surgery with high adherence	28	34.4	18.1–40.8	82.4%	54.2%	34.0%

**Table 8 cancers-17-02321-t008:** Survival results by patient subgroups in glioblastoma (WHO grade 4 glioma).

Characteristic	Subgroup	Number of Patients	One-Year Survival Rate	Statistical Significance
**Gender**				No significant difference
	Male	42	51.9%	
	Female	30	59.1%	
**Age**				No significant difference
	<57 years	35	53.8%	
	≥57 years	37	57.0%	
**Number of Previous Chemotherapy Treatments**				No significant difference
	One treatment	53	58.0%	
	Two or three treatments	13	56.3%	
**Corticosteroid Use**				No significant difference
	Received corticosteroids	31	58.7%	
	Did not receive corticosteroids	41	50.6%	
**Extent of Surgical Intervention**				*p* < 0.001
	Only biopsy	6	1 (16.6%)	
	Gross total resection	28	21 (75.0%)	
	Extensive biopsy/partial resection	38	22 (57.9%)	

**Table 9 cancers-17-02321-t009:** Side effects and management for patients with glioblastoma (WHO grade 4 glioma).

Side Effect Category	Conventional Treatment Side Effects	Observations with Integrative Treatment	Management Strategy
**Radiotherapy Effects**			
Post-radiation edema	Common (G1) and severe (G3) (seen MRI) typically requiring increased steroid use	Decreased in the integrative treatment group	*Boswellia serrata* extract (1.8–2.4 g/day) used to mitigate edema and reduce reliance on glucocorticoids
Post-radiation asthenia	Common complaint affecting quality of life (asked clinically)	Decreased in the integrative treatment group	Integrative treatment protocol (PD + CUR)
**Hematological Parameters**			
Platelet count	Often decreased during chemotherapy, potentially leading to treatment interruption (G2–G3–4)	Normal in the group of 60 highly adherent patients	Regular monitoring and supportive treatments when count was less than 100.000 μL
White blood cell count	Often decreased during chemotherapy, Neutropenia (G1–G3–4)	Normal in the group of 60 highly adherent patients	Tamarix gallica extracts as supportive treatments when (G1–2) present
Red blood cell count	Often decreased during chemotherapy	Normal in the group of 60 highly adherent patients	Tamarix gallica as supportive treatments when red count was under 3000
**Liver Function**			
Liver enzymes	Often elevated during chemotherapy (G2–G3–4)	Not specifically reported	Supportive treatments, including glutathione, when ALT and AST are more than one times the normal level
**Chemotherapy Continuation**			
Treatment interruption	Common due to side effects	All desired cycles completed without interruption in the integrative treatment group	Regular monitoring and prompt management of side effects
**Glucocorticoid Dependence**			
Steroid use for cerebral edema	Common (seen MRI) with associated adverse effects (G1–G3)	Reduced need for steroids	*Boswellia serrata* extract for its anti-edemagenic properties

**Table 10 cancers-17-02321-t010:** Comparing study results with literature data on glioblastoma, WHO grade 4 glioma.

Outcome Measure	Current Study Results	Literature Data	Difference
**Median Overall Survival**			
All patients (n = 72)	13.3 months	12.1–14.6 months	Comparable
Patients who received integrative treatment (n = 65)	16.3 months	12.1–14.6 months	+1.7 to +4.2 months
Patients with high adherence to integrative treatment (n = 60)	25.4 months	12.1–14.6 months	+10.8 to +13.3 months
Patients who underwent complete surgery with high adherence	34.4 months	~18–19 months	+15.4 to +16.4 months
**1-Year Survival Rate**			
All patients (n = 72)	53.1%	~50%	+3.1%
Patients who received integrative treatment (n = 65)	55.4%	~50%	+5.4%
Patients with high adherence to integrative treatment (n = 60)	59.0%	~50%	+9.0%
Patients who underwent complete surgery with high adherence	82.4%	~61–65%	+17.4 to +21.4%
**2-Year Survival Rate**			
Patients who underwent complete surgery with high adherence	54.2%	~25–30%	+24.2 to +29.2%
**5-Year Survival Rate**			
All patients (n = 72)	16.0%	3–7%	+9.0 to +13.0%
Patients with high adherence to integrative treatment (n = 60)	25.0%	3–7%	+18.0 to +22.0%
Patients who underwent complete surgery with high adherence	34.0%	3–7%	+27.0 to +31.0%
**Long-term Survival (>60 months)**			
All patients (n = 72)	12%	<3%	>+9.0%
Patients with high adherence to integrative treatment (n = 60)	16%	<3%	>+13.0%

## Data Availability

The original contributions presented in this study are included in the article. Further inquiries can be directed to the corresponding author.

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
