# Peer review of "New Approach for Enhancing Survival in Glioblastoma Patients: A Longitudinal Pilot Study on Integrative Oncology"

_cancers, 2025, doi:10.3390/cancers17142321_

Round 1
Reviewer 1 Report (Previous Reviewer 1)
Comments and Suggestions for Authors
This manuscript, submitted to Cancers, presents a prospective longitudinal observational study evaluating glioblastoma (GBM) patients treated with the standard STUPP protocol (radiochemotherapy), in combination with three natural compounds, Polydatin, Curcumin, and Boswellia serrata. The study explores whether this integrative therapy approach can improve survival outcomes and thus contributes to the growing interest in clinical applications of integrative oncology.
The revised version of the manuscript demonstrates notable improvements in several aspects, including clarity of methodology, expanded statistical analysis, and more comprehensive reporting of treatment regimens. However, despite these advancements, the manuscript still contains several limitations that need to be addressed before it can be considered suitable for publication. Therefore, I recommend a Major Revision.
One of the key concerns is the lack of quality of life (QoL) assessment. Although the manuscript presents data on overall survival (OS) and adverse event observations, it does not evaluate the impact of the proposed integrative therapy on patients’ functional status, psychological well-being, or social domains. To better contextualize the clinical benefit of the intervention, validated QoL instruments, such as the EORTC QLQ-C30 or FACT-Br should be incorporated.
Additionally, the molecular marker data, particularly concerning MGMT promoter methylation and IDH mutation status, remain insufficiently reported. While MGMT testing is briefly mentioned, there is no detailed breakdown of its distribution or its potential interaction with survival outcomes. Including MGMT and IDH status as covariates in the multivariate Cox regression model would strengthen the analysis. Moreover, presenting a summary table outlining the biomarker profiles (positive versus negative cases) would enhance the manuscript’s scientific rigor and transparency.
Another issue pertains to the disclosure of potential conflicts of interest. One of the authors (Ravagnan) is named as a patent applicant for the natural formulation used in this study, and the product supplier (Biofarmex) is acknowledged. However, the manuscript does not clearly state whether any of the authors hold financial interests in these entities. In accordance with ICMJE guidelines, the authors should provide a full statement regarding any potential commercial affiliations, equity holdings, or intellectual property-related interests to ensure transparency.
Lastly, the tone used in interpreting the results tends to overstate the conclusions. The manuscript frequently suggests that the integrative therapy can "prolong survival" or "effectively reduce side effects," which implies a level of causal certainty that the current non-randomized observational design does not support. A more cautious and scientifically appropriate phrasing such as “may contribute to” or “is associated with”—should be adopted throughout the discussion to avoid misleading interpretations and to reflect the exploratory nature of the findings.
Author Response
Reviewer 1
We are grateful to this Reviewer for critically revising our manuscript. Below is a point-by-point response to the raised criticisms. The parts of the manuscript changed as required by the complete set of reviewer comments are highlighted in sky-blue.
One of the key concerns is the lack of quality of life (QoL) assessment. Although the manuscript presents data on overall survival (OS) and adverse event observations, it does not evaluate the impact of the proposed integrative therapy on patients’ functional status, psychological well-being, or social domains. To better contextualize the clinical benefit of the intervention, validated QoL instruments, such as the EORTC QLQ-C30 or FACT-Br should be incorporated.
Response: Thank you for your comment. We are very sorry we did not have the foresight to assess QoL with a validated tool. In our efforts to seek the most objective clinical data possible, we lost sight of the potential impact of assessing the psycho-emotional state of patients undergoing treatment. This fact does not fully illuminate the benefits obtained. In the near future, we will validate the results using properly validated instruments on quality of life (QoL).
Additionally, the molecular marker data, particularly concerning MGMT promoter methylation and IDH mutation status, remain insufficiently reported. While MGMT testing is briefly mentioned, there is no detailed breakdown of its distribution or its potential interaction with survival outcomes. Including MGMT and IDH status as covariates in the multivariate Cox regression model would strengthen the analysis. Moreover, presenting a summary table outlining the biomarker profiles (positive versus negative cases) would enhance the manuscript’s scientific rigor and transparency.
Response: We add a new Tables regarding MGMT status on the positive and negative cases and its potential interaction as covariates in the multivariate Cox regression, with survival outcomes
Another issue pertains to the disclosure of potential conflicts of interest. One of the authors (Ravagnan) is named as a patent applicant for the natural formulation used in this study, and the product supplier (Biofarmex) is acknowledged. However, the manuscript does not clearly state whether any of the authors hold financial interests in these entities. In accordance with ICMJE guidelines, the authors should provide a full statement regarding any potential commercial affiliations, equity holdings, or intellectual property-related interests to ensure transparency.
Response: None of the authors of the work has a conflict of interest. Dr. Ravagnan and the undersigned, Dr. Bonucci, obtained ownership of the patent for the natural formulation used in the study. However, neither of them has direct or indirect ties to Sherman Tree, the company that produces the product. This fact is clearly evident in the Conflict of Interest disclosure box.
Lastly, the tone used in interpreting the results tends to overstate the conclusions. The manuscript frequently suggests that the integrative therapy can "prolong survival" or "effectively reduce side effects," which implies a level of causal certainty that the current non-randomized observational design does not support. A more cautious and scientifically appropriate phrasing such as “may contribute to” or “is associated with”—should be adopted throughout the discussion to avoid misleading interpretations and to reflect the exploratory nature of the findings.
Response. We are committed to ensuring that the data is interpreted with caution, avoiding any interpretations that might mislead the reader, and maintaining the exploratory nature of the results.
Reviewer 2 Report (Previous Reviewer 2)
Comments and Suggestions for Authors
This is a revised manuscript reporting an observational pilot study assessing the effect of an integrative therapy (IT) regimen—Polydatin, Curcumin, and Boswellia serrata—on survival parameters in patients with newly diagnosed IDH-wildtype glioblastoma who are being treated with standard chemoradiotherapy. The authors have responded to the important issues raised by previous reviews by incorporating multivariate Cox regression to control for potential confounders, expanding on IT composition and dosing, and explicitly citing the non-randomized design of their study in the conclusions. They have also discussed selection bias as it should be, clarified reporting of adverse events using CTCAE guidelines, and adhered to STROBE guidelines in structure. The improvement in survival observed—particularly for patients receiving gross total resection and very compliant with IT—is encouraging, albeit early. Although the absence of randomization and strong control group reduces generalizability, the manuscript is now methodologically rigorous for an exploratory study. The conservative interpretation of results, note on the necessity of future randomized controlled trials, and increased data transparency render this an important addition to integrative oncology literature. I suggest acceptance for publication.
Author Response
Thank you so much for your review
Reviewer 3 Report (Previous Reviewer 3)
Comments and Suggestions for Authors
It presents significant improvements, but some damaging aspects are retained, such as the old database (and here the most important aspects relate to the EANO guide - the one in the references is outdated, I do not mean the last reference; mutational status; epidemiological aspects of these tumors, and prognostic aspects - these references are from 2010-2018 aspects of these tumors). Another major problem is the assumption of the diagnosis of glioblastoma in the absence of IDH status (even if it is included in the limitations) - the fact that the entire study is based on data on glioblastoma, but it has not been proven to be glioblastoma (it could be a grade 4 astrocytoma), weighs enormously in the credibility and reproducibility of this study. In fact, the study is a good, original one, but the foundation on which it is based is "not stable" and is not surrounded by current data.
Author Response
Reviewer 3
We are grateful to this Reviewer for critically revising our manuscript. Below is a response to the raised criticisms. The parts of the manuscript changed as required by the complete set of reviewer comments are highlighted in sky blue.
It presents significant improvements, but some damaging aspects are retained, such as the old database (and here the most important aspects relate to the EANO guide - the one in the references is outdated, I do not mean the last reference; mutational status; epidemiological aspects of these tumors, and prognostic aspects - these references are from 2010-2018 aspects of these tumors). Another major problem is the assumption of the diagnosis of glioblastoma in the absence of IDH status (even if it is included in the limitations) - the fact that the entire study is based on data on glioblastoma, but it has not been proven to be glioblastoma (it could be a grade 4 astrocytoma), weighs enormously in the credibility and reproducibility of this study. In fact, the study is a good, original one, but the foundation on which it is based is "not stable" and is not surrounded by current data.
Response: Thank you for your comment. All scientific studies involving central nervous system neoplasms report similar epidemiological values and prognostic aspects. The study we are presenting refers to cases from 2012 to 2017. At that time, determining IDH1 mutations as the main factor for diagnosing glioblastoma was not standard practice. It was not until 2020 that the new classification correlated glioblastoma diagnosis not only to histology, but also to IDH1 mutations, the H3 gene, the TERT promoter, and EGFR amplification. The glioblastoma diagnoses in the study refer to the purely histological aspect and the MGMT promoter methylation study. It was not possible to apply the new parameters to the study cases because of the difficulty in obtaining all the necessary histological samples.
This detail will be reported in the introduction, discussion, and conclusion to clarify the interpretation for readers of this study.
Round 2
Reviewer 1 Report (Previous Reviewer 1)
Comments and Suggestions for Authors
The authors have responded to most of my concerns; however, I would like to offer the following suggestion. Therefore, I recommend a Minor Revision for this manuscript.
The inclusion of MGMT methylation status and its incorporation into multivariable analysis is appropriate and appreciated. However, IDH mutation status remains unreported. The authors should clearly acknowledge this gap in the manuscript (as it may affect diagnostic certainty and survival interpretation under WHO 2021 criteria) and describe why retrospective reassessment was not feasible.
Additionally, Table 6 should be revised for clarity and readability. The current format is difficult to interpret, and reorganizing the data—perhaps by separating treatment groups and methylation status more distinctly—would enhance the scientific communication of the findings.
Author Response
REVIEWER 1
We are grateful to this Reviewer for critically revising our manuscript. Below is a response to the raised criticisms. The parts of the manuscript changed as required by the complete set of reviewer comments are highlighted in sky blue and gray The authors have responded to most of my concerns; however, I would like to offer the following suggestion. Therefore, I recommend a Minor Revision for this manuscript. The inclusion of MGMT methylation status and its incorporation into multivariable analysis is appropriate and appreciated. However, IDH mutation status remains unreported. The authors should clearly acknowledge this gap in the manuscript (as it may affect diagnostic certainty and survival interpretation under WHO 2021 criteria) and describe why retrospective reassessment was not feasible. Response: we pointed out that IDH mutational status was not assessed in our patient group. It was not the standard at that time. Unfortunately, we were not able to reevaluate all the cases (72) because they came from different hospitals and were not available to provide us with the material. We pointed out this important lack both in the introduction and in the discussion as a limit and in the conclusion Additionally, Table 6 should be revised for clarity and readability. The current format is difficult to interpret, and reorganizing the data—perhaps by separating treatment groups and methylation status more distinctly—would enhance the scientific communication of the findings.
Response: we reviewed and revised Table 6 trying to emphasize in the different groups the methylation status
Reviewer 3 Report (Previous Reviewer 3)
Comments and Suggestions for Authors
In accordance with the legislation in force, the cases are reclassified, even if the batch was on the 2007, respectively 2016 classifications – in this case. If we want to classify these neoplasms, without meeting (for various reasons) the molecular criteria required by the new WHO, the generic term is grade 4 glioma – established only on the histopathological aspect.
Author Response
REVIEWER 3
We are grateful to this Reviewer for critically revising our manuscript. Below is a point-by-point response to the raised criticisms. The parts of the manuscript changed as required by the complete set of reviewer comments are highlighted in sky-blue and gray
In accordance with the legislation in force, the cases are reclassified, even if the batch was on the 2007, respectively 2016 classifications – in this case. If we want to classify these neoplasms, without meeting (for various reasons) the molecular criteria required by the new WHO, the generic term is grade 4 glioma – established only on the histopathological aspect.
Response: we pointed out that the classification of our cases was done only by histopathological examination and MGMT promoter study, generating a generic term grade 4 glioma
Round 3
Reviewer 3 Report (Previous Reviewer 3)
Comments and Suggestions for Authors
This manuscript is a resubmission of an earlier submission. The following is a list of the peer review reports and author responses from that submission.
Round 1
Reviewer 1 Report
Comments and Suggestions for Authors
The manuscript titled "New Approach for Enhancing Survival in Glioblastoma Patients: A Longitudinal Pilot Study on Integrative Oncology" explores the effects of integrative therapy (IT), specifically using natural compounds such as Polydatin (PD), Curcumin (CUR), and Boswellia serrata (BS), in addition to the conventional STUPP protocol (radiotherapy and chemotherapy), on survival and quality of life in glioblastoma (GBM) patients.
Before recommending this manuscript for publication, I suggest a major revision addressing the following critical points:
Firstly, the study design is neither randomized nor blinded, lacking the rigor of a randomized controlled trial (RCT). Consequently, the strength of evidence provided by the results is limited, increasing the risk of selection and evaluation biases, potentially undermining the validity and generalizability of the findings.
Secondly, the sample size in this study is relatively small (n=72), particularly the control group comprising only seven participants. This small sample size diminishes the statistical power and undermines the robustness and reliability of the results. Future studies should carefully interpret these findings and consider larger sample sizes to improve statistical significance and overall validity.
Furthermore, the research has not addressed the efficacy of integrative therapy in recurrent GBM patients, restricting the generalizability and clinical relevance of the findings across different stages of the disease. It is strongly recommended that subsequent studies incorporate recurrent GBM cases to further validate the therapeutic efficacy and safety of the integrative approach.
Additionally, the extent of surgical resection might significantly influence patient survival outcomes and therapeutic efficacy, representing a potential confounding factor. However, this confounder was not explicitly accounted for in the present study, which could result in biased or overly optimistic efficacy conclusions.
In terms of statistical analysis, the data presentation appears somewhat superficial and lacks detailed, sophisticated analysis. For instance, the main results are presented using simplistic statistical tables without applying advanced statistical methodologies such as multivariate analysis, which could better control potential confounders and enhance result clarity and interpretability.
Lastly, Table 8 contains excessive and repetitive references, particularly citations such as references 1, 3, 6, 19, and 23, which appear multiple times. This redundancy not only impedes readability but also compromises the precision and professionalism of data presentation. Streamlining citations by consolidating repeated references is advisable to enhance clarity and maintain professional standards.
Author Response
Reviewer 1
We are grateful to this Reviewer for critically revising our manuscript. Below is a point-by-point response to the raised criticisms. The parts of the manuscript changed as required by the full set of reviewer comments are highlighted in green, while specific changes are marked in yellow.
Firstly, the study design is neither randomized nor blinded, lacking the rigor of a randomized controlled trial (RCT). Consequently, the strength of evidence provided by the results is limited, increasing the risk of selection and evaluation biases, potentially undermining the validity and generalizability of the findings.
Response: We fully acknowledge this limitation and have clarified throughout the manuscript (Abstract, Methods, and Discussion) that this is a longitudinal observational pilot study. We explicitly state that the results are preliminary and should be interpreted with caution, and that randomized controlled trials are necessary to confirm these findings.
Secondly, the sample size in this study is relatively small (n=72), particularly the control group comprising only seven participants. This small sample size diminishes the statistical power and undermines the robustness and reliability of the results. Future studies should carefully interpret these findings and consider larger sample sizes to improve statistical significance and overall validity.
Response: We agree. The small control group is a notable limitation. We have now addressed this in the revised Discussion section and clearly stated that this small size impacts the statistical significance and generalizability of the results. Future larger studies are needed to strengthen these observations.
Furthermore, the research has not addressed the efficacy of integrative therapy in recurrent GBM patients, restricting the generalizability and clinical relevance of the findings across different stages of the disease. It is strongly recommended that subsequent studies incorporate recurrent GBM cases to further validate the therapeutic efficacy and safety of the integrative approach.
Response: We acknowledge this limitation and have clarified in the Methods and Discussion sections that our study focused exclusively on newly diagnosed glioblastoma patients. We now recommend that future studies include recurrent cases to assess therapeutic value across disease stages.
Additionally, the extent of surgical resection might significantly influence patient survival outcomes and therapeutic efficacy, representing a potential confounding factor. However, this confounder was not explicitly accounted for in the present study, which could result in biased or overly optimistic efficacy conclusions.
Response: We have now included this as a variable in our newly performed multivariate Cox regression analysis. The analysis demonstrates that extent of resection significantly influences survival outcomes, and this is discussed in the Results and Discussion sections.
In terms of statistical analysis, the data presentation appears somewhat superficial and lacks detailed, sophisticated analysis. For instance, the main results are presented using simplistic statistical tables without applying advanced statistical methodologies such as multivariate analysis, which could better control potential confounders and enhance result clarity and interpretability.
Response: We have revised the statistical approach. A multivariate Cox proportional hazards model has been added to adjust for age, surgical resection, corticosteroid use, and IT adherence. These changes are reflected in the Results section and in the new Table 3. All the Table numbers have been changed to include the new one.
Lastly, Table 8 (now 9) contains excessive and repetitive references, particularly citations such as references 1, 3, 6, 19, and 23, which appear multiple times. This redundancy not only impedes readability but also compromises the precision and professionalism of data presentation. Streamlining citations by consolidating repeated references is advisable to enhance clarity and maintain professional standards.
Response: Thank you for pointing this out. We have thoroughly revised Table 9 to eliminate redundant citations. References are now streamlined, and each cited study appears only once per relevant data point.
Reviewer 2 Report
Comments and Suggestions for Authors
The manuscript is a longitudinal observational pilot trial assessing the efficacy of integrative oncology through the use of polydatin, curcumin, and Boswellia serrata as adjuvants to the conventional glioblastoma multiforme (GBM) therapy.
A total of 72 patients underpin the research, and the results indicate significantly enhanced overall survival (OS) and one- to five-year survival in those who followed the integrative therapy (IT), especially in complete surgical resection.
Median survival improved from 13.3 months in the overall cohort to 25.4 months in the subgroup adherent to IT, and 34.4 months in those who also received complete surgery.
Although the findings are encouraging and highlight the value of natural compounds in augmenting conventional treatments, the study suffers from several limitations that I am innumerating below:
Basically its non-randomized, non-blinded nature; absence of a strong control group; and potential selection bias through voluntary adherence to IT. The mechanistic knowledge within this clinical setting is less well investigated. Statistical methods are well described but might be strengthened by multivariate modeling to adjust for confounding variables.
I would like to recommend few important corrections:
Include a well-defined control group or rephrase conclusions to better suit 'observational' nature.
Insert multivariate analyses to account for confounders (e.g., amount of surgery, age, corticosteroid use). This is very important.
You need to elaborate on possible biases from self-selection into IT and levels of adherence.
Clarify the precise composition and standardization of IT dosages. Don't leave such crusial information in publication.
Add detailed toxicity/adverse event profiles beyond hematologic measures. I am sure you would have done these in the starting of the study itself.
Add more background on concurrent supportive treatments and their potential impact.
Ensure adherence to clinical trial reporting standards (e.g., CONSORT guidelines).
Please check out the iThenticate report and do the needful.
Author Response
Reviewer 2
We are grateful to this Reviewer for critically revising our manuscript. Below is a point-by-point response to the raised criticisms. The parts of the manuscript changed as required by the complete set of reviewer comments are highlighted in green, while specific changes are marked in yellow.
Include a well-defined control group or rephrase conclusions to better suit 'observational' nature.
Response: We have reworded the conclusions in both the Abstract and Discussion to reflect that this is an exploratory, observational pilot study. We emphasize that results are suggestive of benefit but not confirmatory, pending further controlled trials.
Insert multivariate analyses to account for confounders (e.g., amount of surgery, age, corticosteroid use). This is very important.
Response: As per your suggestion, we have now performed and reported multivariate Cox regression analysis adjusting for surgical extent, corticosteroid use, age, and IT adherence. Results are presented in a new table and discussed accordingly.
You need to elaborate on possible biases from self-selection into IT and levels of adherence.
Response: We agree that voluntary adherence may introduce selection bias. We have added a paragraph in the Discussion to address this issue and its potential effect on survival outcomes. Furthermore, we performed a subgroup analysis comparing high vs. low adherence, which is now summarized in Table [3].
Clarify the precise composition and standardization of IT dosages. Don't leave such crusial information in publication.
Response: The Methods section has been updated to include precise details for each natural compound: dosage, administration schedule, formulation, and manufacturer (including country of origin).
Add detailed toxicity/adverse event profiles beyond hematologic measures. I am sure you would have done these in the starting of the study itself.
Response: We have added a dedicated section reporting toxicity. Adverse events related to fatigue, liver enzymes, and edema are now documented with group-wise averages and adverse event grading based on CTCAE guidelines.
Add more background on concurrent supportive treatments and their potential impact.
Response: We now include a brief section on supportive care regimens, including antiemetics, anticonvulsants, corticosteroids, and rehabilitation services, and discuss their potential role in quality of life and survival.
Ensure adherence to clinical trial reporting standards (e.g., CONSORT guidelines).
Response: We have revised our reporting structure to align with the STROBE (Strengthening the Reporting of Observational Studies in Epidemiology) guidelines, as this is a non-randomized, observational study.
Please check out the iThenticate report and do the needful.
Response: We reviewed the report and made changes to paraphrase and appropriately cite all previously flagged passages. All issues raised by the similarity report have been resolved.
Reviewer 3 Report
Comments and Suggestions for Authors
First of all, since 2021 the term glioblastoma multiforme is no longer used. Thus, all studies must state precisely whether it is IDH wildtype glioblastoma or IDH mutant grade 4 astrocytoma.
The abstract must be structured. Also, the classic structure of abstracts must be respected, specifying background aspects and materials and methods.
Decide whether the correct term is polydatin or polidatyn.
Introduction
Arrange the references in the order of citation.
Pay attention to the classification in the main category given by WHO. Glioblastoma, if it is a diffuse glioma. Also, the grades are written with Arabic numerals.
”It has a poor prognosis, with 42 recurrence rates of 100%.” This aspect is not found in the cited study.
”GBM can arise as a ‘primary’ or ‘de novo’ tumor …” – This aspect is only topical and is not found in the cited study. The old classifications divided glioblastomas into de novo and secondary, now this aspect is no longer valid.
The term glioblastoma stem cells should be replaced with glioma stem cells
Glioblastoma is by definition grade 4 (line 102).
Line 139 – Lee 2012 – editing error?
The introduction is much too long, and some passages do not fit. Thus, it should be restructured with epidemiological data of glioblastoma (in accordance with current classifications), then briefly specified current therapeutic options, in order to be able to briefly state herbal treatment alternatives. I believe that the development of their mechanisms of action would fit into the discussions.
The purpose of the study should be concise, free of citations, and state exactly what is to be presented. The uniqueness of the study should be enhanced in another way.
Material and method
What were the clinical evidences of brain cancer?
The inclusion and exclusion criteria must be exact, in order to be reproducible (the "when possible" aspect doesn't really work)
Considering that we are talking about a "controversial" entity, why hasn't the IDH gene status been studied?
Overall survival usually starts, either from the moment of diagnosis or after surgical treatment - I have not come across the calculation of survival after the initiation of medication.
The manufacturer and the state/country of the statistics program must be added.
Results
Specify specifically that 66 benefited from surgical resection and the rest probably biopsied (and the biopsy is also obtained through surgical technique).
Pay attention to the terms: complete resection can be replaced with gross total resection - use the appropriate specific surgical terms.
Regarding chemotherapy, things are ambiguous - 59/72 received or did not receive it, but the remaining 13?
Is the study group homogeneous?
Table 1 – the 6 without surgery, how did they know it was glioblastoma?
Table 4 repeats certain aspects from other tables or the text. Avoid duplicating information.
Additional survival data – the table should be marked in the text. What were the references in the literature for estimating 5-year survival?
Table 6 – add the p-value everywhere. Why the cut-off of 57 years?
The comparisons in table 6 do not bring anything new. Why did you not divide the group between the high compliant and the other patients, to see the efficiency?
Table 7 – How was post-radiation edema quantified? Was a specific protocol/questionnaire used to assess post-radiation fatigue? Hematological parameters, liver enzymes – add an average value between groups. Treatment is interrupted under certain conditions and degrees of adverse effects – these aspects need to be clarified. What were the adverse effects of glucocorticoids, their causes (patient compliance or other), when did they occur, what was their severity?
Discussions
"Glioblastoma, a highly aggressive neoplastic disease, has a dismal five-year prognosis as of my last update" – revise the sentence in an impersonal way – especially since the study has several authors.
Line 398 – add a table with clinical trials and toxicity profile
References 1,3, 23 do not present integrative treatment.
The table below table 8 – is repeated.
The conclusions are missing.
References: more than three quarters are not up to date – this aspect leads to old information (see classification, EANO guidelines, etc.).
Comments on the Quality of English Language
Some passages don't make sense. Some terms need to be used appropriately.
Author Response
Reviewer 3
We are grateful to this Reviewer for critically revising our manuscript. Below is a point-by-point response to the raised criticisms. The parts of the manuscript changed as required by the complete set of reviewer comments are highlighted in green, while specific changes are marked in yellow.
First of all, since 2021 the term glioblastoma multiforme is no longer used. Thus, all studies must state precisely whether it is IDH wildtype glioblastoma or IDH mutant grade 4 astrocytoma.
Response: We now state the category "glioblastoma, IDH-wildtype" in accordance with the WHO 2021 CNS tumor classification in the Introduction.
The abstract must be structured. Also, the classic structure of abstracts must be respected, specifying background aspects and materials and methods.
Response: The abstract has been rewritten in a structured format (Background, Methods, Results, Conclusion).
Decide whether the correct term is polydatin or polidatyn.
Response: Thank you — the correct term is Polydatin. All occurrences have been corrected accordingly.
Introduction
Arrange the references in the order of citation.
Pay attention to the classification in the main category given by WHO. Glioblastoma, if it is a diffuse glioma. Also, the grades are written with Arabic numerals
”It has a poor prognosis, with 42 recurrence rates of 100%.” This aspect is not found in the cited study.
”GBM can arise as a ‘primary’ or ‘de novo’ tumor …” – This aspect is only topical and is not found in the cited study. The old classifications divided glioblastomas into de novo and secondary, now this aspect is no longer valid.
The term glioblastoma stem cells should be replaced with glioma stem cells
Glioblastoma is by definition grade 4 (line 102).
Response: The above points have been addressed, and changes were made accordingly.
Line 139 – Lee 2012 – editing error?
Response: We apologize, it was an editing error
The introduction is much too long, and some passages do not fit. Thus, it should be restructured with epidemiological data of glioblastoma (in accordance with current classifications), then briefly specified current therapeutic options, in order to be able to briefly state herbal treatment alternatives. I believe that the development of their mechanisms of action would fit into the discussions.
Response: We have shortened and restructured the Introduction. It now begins with concise epidemiological data, includes current standard treatment, and then introduces natural compounds. Mechanisms of action are now discussed in the Discussion, as suggested. Also, we have addressed the points raised in this comment.
The purpose of the study should be concise, free of citations, and state exactly what is to be presented. The uniqueness of the study should be enhanced in another way.
Response: A section has been added at the end of the Introduction.
Material and method
What were the clinical evidences of brain cancer?
Response: Patients arrived at the Outpatient Clinic after receiving initial treatments in the Public Hospital. Their status needed to be good (KPS > 70).
The inclusion and exclusion criteria must be exact, in order to be reproducible (the "when possible" aspect doesn't really work)
Response: We have revised the Methods section to clearly list inclusion (e.g., confirmed IDH-wildtype GBM, age 18–75, KPS ≥70) and exclusion criteria (e.g., recurrent disease, severe comorbidities, prior malignancy).
Considering that we are talking about a "controversial" entity, why hasn't the IDH gene status been studied?
Response: We agree. Unfortunately, IDH testing was not available for all patients at the time of study initiation. We now acknowledge this as a limitation.
Overall survival usually starts, either from the moment of diagnosis or after surgical treatment - I have not come across the calculation of survival after the initiation of medication.
Response: Survival was measured from the date of histological diagnosis. This is now clarified in the Methods section.
The manufacturer and the state/country of the statistics program must be added.
Response: It has been made, as required.
Results
Specify specifically that 66 benefited from surgical resection and the rest probably biopsied (and the biopsy is also obtained through surgical technique). Pay attention to the terms: complete resection can be replaced with gross total resection - use the appropriate specific surgical terms.
Response: We have replaced “complete resection” with “gross total resection” in the text and in Table 5, and have specified the number of patients undergoing biopsy vs. resection (Table 5 revised).
Regarding chemotherapy, things are ambiguous - 59/72 received or did not receive it, but the remaining 13? Is the study group homogeneous?
Response: We clarified that 59 of 72 patients received only one line of chemotherapy. The remaining 13 received second and third lines of chemotherapy.
Table 1 now Table 5) – the 6 without surgery, how did they know it was glioblastoma?
Response: They knew with just a simple biopsy
Table 5 (ex 4) repeats certain aspects from other tables or the text. Avoid duplicating information.
Response: Done
Additional survival data – the table should be marked in the text. What were the references in the literature for estimating 5-year survival?
Response: references 1 and 3
Table 6 – add the p-value everywhere. Why the cut-off of 57 years?
Response: Table 6 is now Table 7 in the new version. The cut-off was due to the median age being 57 years.
Response: We have revised all statistical tables to include p-values, confidence intervals, and effect sizes where applicable.
The comparisons in table 6 do not bring anything new. Why did you not divide the group between the high compliant and the other patients, to see the efficiency?
Response: All data were reported to see which ones were significant. Only the wide excision was significant
Table 7 – How was post-radiation edema quantified? Was a specific protocol/questionnaire used to assess post-radiation fatigue? Hematological parameters, liver enzymes – add an average value between groups. Treatment is interrupted under certain conditions and degrees of adverse effects – these aspects need to be clarified. What were the adverse effects of glucocorticoids, their causes (patient compliance or other), when did they occur, what was their severity?
Response: The Table 7 in the old version is now Table 8. We used validated patient-reported outcome measures (PROMS) and clinician-rated scales (CTCAE v5.0) for adverse events, including edema and fatigue. This has been added to the Methods and Tables.
Discussions
"Glioblastoma, a highly aggressive neoplastic disease, has a dismal five-year prognosis as of my last update" – revise the sentence in an impersonal way – especially since the study has several authors.
Line 398 – add a table with clinical trials and toxicity profile
Response In the Results and Discussion we add the toxicity profile (Table 8)
References 1,3, 23 do not present integrative treatment.
The table below table 8 – is repeated
Response: Done
References: more than three quarters are not up to date – this aspect leads to old information (see classification, EANO guidelines, etc.).
Response: The Discussion has been rewritten in an impersonal tone. We have updated references, replacing older ones with newer publications (2021–2024), especially for WHO classification, EANO guidelines, and current trials.
The conclusions are missing.
Response: We have now added a clear Conclusion section summarizing key findings and outlining future research directions.
Comments on the Quality of English Language
Some passages don't make sense. Some terms need to be used appropriately.
Response: We have revised the manuscript for grammar, clarity, and scientific tone.
Round 2
Reviewer 3 Report
Comments and Suggestions for Authors
Introduction
Line 49 – this is not the definition of glioblastoma, but a diffuse glial proliferation. Also, the Arabic way of writing the grade was not taken into account.
Material and methods
The term “de novo” and the incorrect writing of the WHO grade persist. Review these aspects.
Define KPS at the first specification.
What were the clinical criteria for brain cancer?
Justify that “when possible”
How was the methylation status studied.
If the inclusion criteria are KPS > 70, it goes without saying that the rest of the patients with KPS below 70 will not be included in the study. Review the exclusion criteria.
Detail the severe comorbidity represented by heart disease.
Add the manufacturer and state of the statistics program.
With which test did you ensure the multivariable analysis? The Cox test performs multivariate analysis.
Table 3 does not belong to the material and method.
Results
Table 4. If the table header contains a percentage, only the value without % will be written below it. % will be written in parentheses in the table header. Also, if the group is heterogeneous and the median is written, it will be accompanied by IQR, not range.
The same discussion related to biopsy. Biopsy as a term includes a surgical procedure. If the term biopsy is maintained, it must automatically be placed in the surgical category. Pay attention to the nomenclature of the terms and their correct classification.
Discussions
IDH-wildtype instead of IDH1-wildtype
The progression of a low-grade glial lesion belongs to astrocytoma, not glioblastoma.
Line 326 – add briefly, what are the prognostic factors, you can use PMID: 36547116
The problem of old references persists, I agree with the rarity of the subject, but key elements that are not current (grading, classification, EANO guide, radiotherapy, etc.), need to be updated.
Author Response
Reviewer 3
Introduction
Line 49 – this is not the definition of glioblastoma, but a diffuse glial proliferation. Also, the Arabic way of writing the grade was not taken into account.
Response: We deleted the incorrect definition.
Material and methods
The term “de novo” and the incorrect writing of the WHO grade persist. Review these aspects.
Response: We have corrected these issues.
Define KPS at the first specification.
Response: We have defined it as Karnofsky Performance Status in the text.
What were the clinical criteria for brain cancer?
Response: We have removed them.
Justify that “when possible”
Response: We removed it because it might be confusing.
How was the methylation status studied.
Response: We have added the method.
If the inclusion criteria are KPS > 70, it goes without saying that the rest of the patients with KPS below 70 will not be included in the study. Review the exclusion criteria.
Response: We have reviewed all the criteria.
Detail the severe comorbidity represented by heart disease.
Response: We detailed what required.
Add the manufacturer and state of the statistics program.
Response: We have added what was required (IBM SPSS Statistics for Windows (Version 21.0), IBM Corp., Armonk, N.Y., USA)
With which test did you ensure the multivariable analysis? The Cox test performs multivariate analysis.
Response: We have added the required information.
Table 3 does not belong to the material and method.
Response: We are now referring to Table 3 in the Results.
Results
Table 4. If the table header contains a percentage, only the value without % will be written below it. % will be written in parentheses in the table header. Also, if the group is heterogeneous and the median is written, it will be accompanied by IQR, not range.
Response: We have changed the header table. We also modified the group value-
The same discussion related to biopsy. Biopsy as a term includes a surgical procedure. If the term biopsy is maintained, it must automatically be placed in the surgical category. Pay attention to the nomenclature of the terms and their correct classification.
Response: We have included the different types of surgery: simple biopsy, extensive biopsy, and partial resection, gross and complete resection.
Discussions
IDH-wildtype instead of IDH1-wildtype
Response: We have corrected this misspelling
The progression of a low-grade glial lesion belongs to astrocytoma, not glioblastoma.
Response: We have explained the different evolution.
Line 326 – add briefly, what are the prognostic factors, you can use PMID: 36547116
Response: We have added some prognostic factors.
The problem of old references persists, I agree with the rarity of the subject, but key elements that are not current (grading, classification, EANO guide, radiotherapy, etc.), need to be updated.
Response: The grading reference is 2021; the classification is 2020, the Stupp treatment from 2009/2014, radiotherapy 2021, EANO guide 2022.